# Peptide-Conjugated Phosphorodiamidate Morpholino Oligomers for In Situ Live-Cell Molecular Imaging of Dengue Virus Replication

**DOI:** 10.3390/ijms21239260

**Published:** 2020-12-04

**Authors:** Carla Bianca Luena Victorio, Wisna Novera, Jing Yang Tham, Satoru Watanabe, Subhash G. Vasudevan, Ann-Marie Chacko

**Affiliations:** 1Laboratory for Translational and Molecular Imaging, Cancer and Stem Cell Biology Programme, Duke-NUS Medical School, Singapore 169857, Singapore; carla-bianca.victorio@duke-nus.edu.sg (C.B.L.V.); wisna.novera@duke-nus.edu.sg (W.N.); jingyang.tham@duke-nus.edu.sg (J.Y.T.); 2Programme in Emerging Infectious Diseases, Duke-NUS Medical School, Singapore 169857, Singapore; satoru.watanabe@duke-nus.edu.sg (S.W.); subhash.vasudevan@duke-nus.edu.sg (S.G.V.)

**Keywords:** in situ molecular imaging, live-cell fluorescence imaging, antisense oligonucleotides, virus tracking, fluorescence imaging probes

## Abstract

Current methods to detect and monitor pathogens in biological systems are largely limited by the tradeoffs between spatial context and temporal detail. A new generation of molecular tracking that provides both information simultaneously involves in situ detection coupled with non-invasive imaging. An example is antisense imaging that uses antisense oligonucleotide probes complementary to a target nucleotide sequence. In this study, we explored the potential of repurposing antisense oligonucleotides initially developed as antiviral therapeutics as molecular probes for imaging of viral infections in vitro and in vivo. We employed nuclease-resistant phosphorodiamidate synthetic oligonucleotides conjugated with cell-penetrating peptides (i.e., PPMOs) previously established as antivirals for dengue virus serotype-2 (DENV2). As proof of concept, and before further development for preclinical testing, we evaluated its validity as in situ molecular imaging probe for tracking cellular DENV2 infection using live-cell fluorescence imaging. Although the PPMO was designed to specifically target the DENV2 genome, it was unsuitable as in situ molecular imaging probe. This study details our evaluation of the PPMOs to assess specific and sensitive molecular imaging of DENV2 infection and tells a cautionary tale for those exploring antisense oligonucleotides as probes for non-invasive imaging and monitoring of pathogen infections in experimental animal models.

## 1. Introduction

Real-time in situ molecular imaging [1] is a non-invasive imaging technique that surpasses traditional methods of pathogen detection and monitoring by providing both spatial and temporal information. Such methods provide spatial localization of the pathogen that circumvents the problem of spatial heterogeneity inherent in most diseases; and enable real-time monitoring of infection kinetics. These are both crucial in studies elucidating disease pathogenesis and evaluating the therapeutic efficacy of candidate vaccines and antimicrobials. Real-time in situ molecular imaging employs probes that bind to either the genome, transcribed genes, or infection-specific proteins expressed over time in live cells, tissues, or animals [2]. This approach relies on biocompatible, non-toxic pathogen-specific probes detectable in vitro, ex vivo, and/or in vivo by molecular imaging technologies ranging from optical imaging (e.g., fluorescence, and bioluminescence), to magnetic imaging (e.g., magnetic resonance imaging, MRI), and nuclear imaging (e.g., positron emission tomography, PET; single-photon emission computed tomography, SPECT) [3]. PET imaging of [^18^F]fluorodeoxyglucose (FDG) uptake in tissues is currently the most widely used non-invasive method for identifying foci of tissue infection and inflammation in the clinic [4]. FDG is a radioactive analog of glucose actively taken up by tissues with high metabolic rate, such as in the case of inflammation associated with viral infection. FDG-PET had been successfully used by our group to visualize host inflammatory response to acute dengue virus (DENV) infection in a mouse model [5]. It had also been used in imaging disease pathogenesis and/or therapy in experimental animal models with monkeypox virus [6,7], influenza [8], and MERS-CoV coronavirus [9] infections. Despite the potential value of FDG-PET imaging for viral infections, FDG is not pathogen-specific and cannot distinguish different infections in vivo such as DENV and Zika virus (ZIKV), which are our group’s primary interest. An alternative molecular imaging technique that uses pathogen-specific probes is antisense imaging.

Antisense imaging utilizes single-strand synthetic antisense oligonucleotide probes with a sequence complementary to the target nucleic acid of interest (i.e., the sense-strand) and tagged with an imaging contrast agent [10]. Antisense imaging has been widely applied in the preclinical setting for non-invasive molecular imaging of cancer [11,12,13], while there are limited examples for imaging infections [14,15,16]. Antisense imaging has never been translated into clinical research or practice. One of the most commonly used synthetic antisense oligonucleotides are the nuclease-resistant phosphorodiamidate morpholino oligomers (PMO) in which the deoxy/ribose sugars are supplanted with morpholino structures (reviewed in [17]). PMOs do not readily cross the plasma membrane [18] and so are often conjugated with cell-penetrating peptides (CPP) to facilitate intracellular delivery (reviewed in [19,20]). Various CPP-conjugated PMOs (subsequently referred to as PPMOs) have been developed and evaluated as antiviral therapeutics for a number of viral infections [21,22,23,24,25,26], including coronaviruses [27,28], ZIKV [29], and Dengue virus (DENV) [30,31,32,33] (reviewed in [34,35]). These antivirals are reported to inhibit viral replication by sequence-specific Watson–Crick base-pairing of the PPMOs with the target viral genome. Hence, there is potential for repurposing these inherently target-specific PPMO antiviral agents as antisense imaging probes for non-invasive tracking of viral infections in vitro and in vivo.

This study investigates the compatibility of PPMO antiviral agents as antisense imaging probes for real-time in situ molecular detection and monitoring of DENV infection. DENV is a mosquito-transmitted virus, and infection is clinically manifested as a spectrum of symptoms from mild febrile illness to lethal dengue hemorrhagic shock syndrome [36]. We selected from the published literature a PPMO that inhibited DENV serotype-2 (DENV2) replication in monkey kidney (Vero) cells [31], and prolonged survival of infected mice [32]. Originally developed by AVI Biopharma [31,32,37], this 24-mer 5′SL PPMO was designed with 100% complementary sequence to the stem-loop structure (SL) of the 5′ terminus untranslated region of DENV2 genome [30], which is highly conserved among DENV2 strains. Here we evaluated 5′SL PPMO as a live-cell imaging probe of DENV2 replication in two relevant DENV cellular infection models—monkey kidney (Vero) cells and baby hamster kidney (BHK-21) cells. Using fluorescence imaging, we determined whether the 5′SL PPMO spatially and temporally tracks DENV2 cellular infection.

## 2. Results

We evaluated the potential of antiviral 5′SL PPMO (Figure 1A), which was designed to bind the Dengue virus seroptype-2 (DENV2) genome (Appendix A), as an imaging agent for DENV2 cellular infection; and compared its performance as a DENV-specific probe to a non-targeted control (CTRL) PPMO that was designed to not bind to the DENV2 genome (Appendix A). These PPMOs were described as naturally stable and delivered into cells with the aid of a cell-penetrating peptide (RxR)_4_B (Figure 1A) [38]. The 5′SL PPMO sequence was also target-specific and did not target other mRNA sequences in the cell (Appendix A). Moreover, the 5′SL and CTRL PPMOs were labeled with distinct fluorophore tags to facilitate simultaneous imaging in cells (Figure 1A).

To determine the PPMO concentration most suitable for subsequent imaging experiments, we evaluated three different PPMO concentrations (1 µM, 10 µM, and 50 µM) for cytotoxicity and fluorescence imaging contrast. Monkey kidney (Vero) cells were incubated with either PPMO for 10 min and subsequently subjected to live fluorescence imaging. Acute cytotoxicity was observed in cells incubated with 50 µM 5′SL PPMO as early as 2 h after PPMO incubation, but not at lower concentrations (1 or 10 µM) (Figure 1B). Similar phenomena were observed with CTRL PPMO (data not shown). To determine the PPMO concentration providing highest imaging contrast, we quantified the intensity of fluorescence signals contributed by PPMOs in the cytoplasm where viral replication was known to occur. At 2 h after PPMO incubation, we observed fluorescent green (5′SL PPMO) and red (CTRL PPMO) punctae scattered throughout the cytoplasm (Figure 1C), which confirmed cellular entry of these PPMOs. The punctae were attributed to PPMOs concentrated in cellular vesicles, possibly endosomes. Diffuse fluorescence signals were also observed in both the cytoplasm and nucleus (Figure 1C). To quantify these fluorescence signals, the Corrected Total Cell Fluorescence (CTCF) value was determined in individual cells by obtaining the integrated signal density within regions-of-interest (ROIs) drawn around the cytoplasm and subtracting the contribution of background within these ROIs (Figure 1C) (See Methods for full details of image analysis). Cells incubated with 10 µM PPMOs exhibited the highest CTCF values for both PPMOs (Figure 1D) and therefore afforded the best imaging contrast. On the other hand, cells incubated with 1 µM PPMOs exhibited low-intensity signals that were indistinguishable from background (Figure 1C,D). Hence, 10 µM PPMO concentration was selected for subsequent imaging assays due to low toxicity and superior imaging contrast associated with this PPMO concentration.

We next determined whether the 5′SL PPMO specifically tracked DENV2 cellular infection. DENV2-infected Vero and BHK-21 cells were incubated with both PPMOs (10 µM each, 10 min) at 48 h and 72 h post-infection and subsequently immunostained with antibodies targeting double-stranded RNA (dsRNA). DENV replication had been known to occur in membrane-bound multi-protein structures that assemble within vesicle packets and produces double-stranded RNA (dsRNA) as replication intermediates [39]. These vesicle packets were observed as intensely fluorescent punctae around nuclei in both time points assayed (Figure 2A,B). To evaluate whether the 5′SL PPMO colocalized with viral replication vesicle packets, we calculated the Manders split coefficient (tM1) [40] between dsRNA (proxy for viral replication) and either 5′SL or CTRL PPMO. The tM1_dsRNA_ describes the fraction of total intensity from the dsRNA fluorescence channel located in pixels where the intensity from the other fluorescence channel exceeds a threshold automatically determined by the software. The amount of dsRNA fluorescence signals that colocalized with 5′SL PPMO (tM1_dsRNA + 5′SL PPMO_) was similar to the amount of dsRNA fluorescence colocalized with CTRL PPMO (tM1_dsRNA + CTRL PPMO_) in both DENV2-infected Vero and BHK-21 cells regardless of time post-infection (Figure 2C). This indicated that the extent of colocalization of dsRNA with 5′SL PPMO was comparable to the extent of colocalization of dsRNA with CTRL PPMOs. Moreover, the vesicle packets were not selectively enriched with 5′SL PPMOs over time. This was not entirely surprising, since 5′SL and CTRL PPMOs exhibited high colocalization with each other (Figure 2D,E), more than they colocalized with dsRNA. Similar observations were noted in cells infected with ZIKV (Appendix A), where the PPMOs colocalized more with each other than with the ZIKV vesicle packets. These results suggested that the majority of PPMOs had been either occupying the same vesicles or trapped in endosomes and were not found within vesicle packets where DENV2 replication occurs.

To evaluate how the kinetics of PPMO distribution in cells correlated with DENV2 replication, we performed live-cell imaging at various time points of DENV2-infected cells incubated with both PPMOs (10 µM each, 10 min) at 1 h post-infection. Both mock-infected and DENV2-infected cells exhibited peri-nuclear fluorescent punctae contributed by both 5′SL and CTRL PPMOs (Figure 3A), and this distribution was consistent over time (Appendix A). In both mock-infected and DENV2-infected Vero cells, the corrected fluorescence intensity contributed by either 5′SL or CTRL PPMO was constant over time (Figure 3B), suggesting slow clearance of the PPMOs out of the cells. Moreover, the ratio of 5′SL PPMO corrected fluorescence intensity in infected vs. mock-infected cells indicated that the 5′SL PPMO was not selectively enriched in DENV-infected cells at any time point assayed (Figure 3B). In contrast to Vero, both PPMOs cleared out of BHK-21 cells within 48 h regardless of infection status (Figure 3C), and the 5′SL PPMO was slightly enriched in virus-infected vs. mock-infected BHK-21 (CTCF_DENV_/CTCF_MOCK_ > 1) during the first 24 h of infection (Figure 3C). However, the CTRL PPMO was similarly enriched in BHK-21 cells regardless of infection status (Figure 3C). Despite differences in PPMO clearance kinetics in the two cell lines, the DENV2 replication kinetics in both cells were comparable (Figure 3D). Hence, these findings demonstrated that the kinetics of PPMO clearance varied independently of DENV2 replication kinetics in either cell line.

Due to the antiviral nature of 5′SL PPMO, we assessed whether it affected DENV2 infection at the intended imaging dose (10 µM final concentration), which would confound its application as a non-interfering imaging probe for DENV infection. Incubation of Vero cells with 5′SL PPMO (10 min duration) 1 h prior to DENV2 infection resulted in a precipitous drop in virus production to non-detectable levels at 3 days and 6 days post-infection (Appendix A), which corroborated previous observations [31,32]. However, this was not observed in cells inoculated with ZIKV (Appendix A), which confirmed that the antiviral effect was specific to DENV2 infection. Indeed, when we assessed DENV2 replication in Vero cells at various time points post-infection and following short incubation with 5′SL PPMO, we observed a drastic reduction in cellular content of viral RNA (Figure 3E). This confirmed that 5′SL PPMO inhibited DENV2 replication as early as 24 h post-infection. Thus, 5′SL PPMO was deemed unsuitable as antisense imaging probe when used at 10 µM concentration because of its pharmacological effect on the phenomenon it was tracking. Interestingly, pre-treatment with CTRL PPMO also reduced DENV2 production by as much as 50% (*p* < 0.0001) (Appendix A), which contradicted previous reports [30,32]. We speculated that mechanisms other than the canonical Watson–Crick base pairing between the CTRL PPMO and DENV2 genome may have contributed to inhibition of viral replication. One possibility was electrostatic interactions of the CPP with viral proteins or host proteins assembled in the viral replication complex [41].

This study evaluated antiviral PPMOs as potential in situ molecular imaging probes for cellular-based infection assays using DENV2-targeted 5′SL and an unrelated non-targeted CTRL PPMOs. These PPMOs were used as live-cell fluorescence imaging agents to determine whether it tracked spatially and temporally DENV cellular infection in an initial validation test before further development as probes for in vivo PET/SPECT tracking. For the assays, we used Vero and BHK-21 cells—the two most widely used in vitro models of DENV cellular infection [30,32,34]. Based on low cytotoxicity and superior imaging contrast, 10 µM final PPMO concentration was selected as the optimal dose for live-cell fluorescence imaging. From immunofluorescence staining of DENV-infected cells incubated with PPMOs at 48 h or 72 h post-infection, the 5′SL PPMO was determined to not colocalize with viral vesicle packets where viral replication occurs. Instead, the majority of 5′SL PPMOs seemed to occupy the same cellular vesicles as CTRL PPMOs, and this could have been an artifact of the general route of PPMO cellular entry: PPMOs were generally delivered into cells by CPPs that nonspecifically triggered endocytosis. Another possibility was that many of these PPMOs become trapped in endosomes, never reaching their target replicating DENV RNA—a likely scenario that had been previously highlighted as a major barrier for ASOs to become successful cellular therapeutics [20,42]. Though the DENV-specific targeting of these PPMOs could be evaluated independently of cellular entry (e.g., through transient electroporation), this approach would not be translatable to non-invasive preclinical and clinical imaging applications. Hence, future generations of antisense imaging probes would need to include alterations in the CPP to enhance endosomal escape [43] and improve delivery of the probe to the intended target [20]. In addition, the kinetics of PPMO clearance from cells did not coincide with the kinetics of viral replication. Instead, we found the 5′SL PPMO drastically inhibited DENV2 replication in cells at the imaging dose (10 µM). Due to the limited detection sensitivity of optical imaging, it was not possible to evaluate whether using the PPMO at lower concentrations (<10 µM) would better track DENV infection without inhibiting viral replication. This limitation could be circumvented in the future by tagging the PPMOs with probes amenable to more sensitive detection methods, such as PET or SPECT imaging, which could potentially reach optimal imaging dose in the nanomolar concentration range. However, radiolabeling of PPMOs with appropriate PET/SPECT radionuclides includes new chemical modifications which could affect both the chemical integrity, as well as the biological behavior and targeting of the radiolabeled PPMO probe relative to the original PPMO. Hence, radiolabeled PPMO probes will have to go through a similar battery of tests as those described herein to demonstrate their suitability for their intended use.

In conclusion, the potential of 5′SL PPMO as an antisense imaging probe for DENV cellular infection is largely limited by its nonspecific route of intracellular delivery, the need for additional requirement for endosomal escape, and its intrinsic potency in inhibiting viral replication. There is a misconception that because ASOs are designed to be target-specific, then it will specifically track the target (spatially and temporally). Our results demonstrate that this is not the case, and our study serves as a cautionary tale for those exploring the use synthetic ASOs as probes for non-invasive imaging and monitoring of viral infections in experimental animal models. As potential imaging agents, and as candidate therapeutics, several barriers and challenges need to be overcome, which have been discussed thoroughly in past reviews [42,44]. With the increasing global risk of viral pandemics—such as the high transmissibility and virulence of the current SARS-CoV-2 outbreak—pathogen-specific non-invasive imaging probes are highly valued and urgently needed [45]. However, unless the inherent limitations of ASOs are circumvented, antisense imaging can take the back seat for now as we focus our attention on other candidate pathogen-specific imaging probes.

## 3. Materials and Methods

### 3.1. Materials Used in This Study

Vero (ATCC^®^ CCL-81; monkey kidney epithelial cells) and BHK-21 (ATCC^®^ CCL-10; baby hamster kidney epithelial cells) cell lines were purchased from the American Type Tissue Culture Collection (ATCC, Manassas, VA, USA) and confirmed free of Mycoplasma by PCR methods [46] prior to use. Vero cells were cultured in DMEM (Gibco BioSciences, Dublin, Ireland), and BHK-21 cells were cultured in RPMI-1640 (Gibco BioSciences, Dublin, Ireland), and media supplemented with 10% (*v*/*v*) Fetal Bovine Serum (FBS).

The mouse-adapted Dengue virus serotype-2 (DENV2) S221 strain was a gift from Prof. Sujan Shresta (La Jolla Institute for Immunology, La Jolla, CA, USA) [47]. The Zika virus (ZIKV) French Polynesia strain (H/PF/2013) was obtained from the European Virus Archive. Both virus strains were propagated in mosquito (*Aedes albopictus*) C6/36 cell line (ATCC^®^ CRL-1660) prior to use.

The DENV-targeted (5′SL) and non-targeted unrelated control (CTRL) morpholino oligomers (PMOs) conjugated with cell-penetrating peptides (PPMOs) were obtained from Prof. Hong Moulton and Dr. David Stein at Oregon State University (Corvallis, OR, USA). Both PPMOs were synthesized as previously described [37]. Briefly, PPMOs were conjugated at the 3′-end with fluorophores and at the 5′-end with the cell-penetrating peptide (RXR)_4_XB [48], where R = arginine, X = aminohexanoic acid, and B = β-alanine. The DENV-targeted 5′SL PPMO has the following sequence: 5′ GTC GGT CCA CGT AGA CTA ACA ACT 3′ and was designed to bind to the stem-loop (SL) region at the 5′ untranslated region (5′UTR) of DENV2 genomes [30]. The 5′SL PPMO was tagged with carboxyfluorescein (FAM) with excitation and emission peak wavelengths (λ_Ex/Em_) of 495 nm and 519 nm, respectively. The CTRL PPMO has the following sequence: 5′ CCT CTT ACC TCA GTT ACA ATT TAT A 3′ and was tagged with lissamine rhodamine (LRB, λ_Ex/Em_ = 560/583 nm).

### 3.2. Virus Infection and PPMO Incubation

Cells were seeded (2.5 × 10^4^ cells/well) overnight in 24-well *µ*-plates (Ibidi, Munich, Germany) and inoculated with virus at a multiplicity of infection (MOI) of 1 (i.e., 1 plaque-forming unit (pfu) per cell). Cells were incubated with virus in serum-free media (either DMEM or RPMI-1640) for 1 h at 37 °C, 5% CO_2_. Afterwards, the virus inoculum was removed and replaced with PPMO solution in sterile PBS containing both the targeted (5′SL, 10 µM final concentration) and non-targeted control (CTRL, 10 µM final concentration) PPMOs, as well as Hoechst 33,342 (λ_Ex/Em_, 350/461 nm) nuclear counterstain (1 µg/mL). Cells were incubated in the PPMO solution for 10 min at 37 °C, 5% CO_2_ and afterwards washed twice in sterile PBS solution. Cells were finally maintained in culture media supplemented with 2% (*v*/*v*) FBS and incubated at 37 °C, 5% CO_2_. For assays involving determining colocalization of double-stranded RNA (dsRNA) with PPMOs, PPMOs were incubated with cells at 48 h or 72 h post-infection. For live-cell imaging assays at various timepoints and determining viral replication kinetics in the presence of 5′SL PPMO, PPMOs were incubated with cells at 1 h post-infection. After PPMO incubation, cells were washed twice in sterile PBS solution and incubated in fresh culture media prior to live-cell imaging.

### 3.3. Immunofluorescence (IF) Staining

Cells were fixed in 4% paraformaldehyde for 20 min at room temperature (RT) at 48 h or 72 h post-infection and subsequent PPMO addition. Cells were permeabilized in 0.3% Triton X-100 (15 min, RT) and blocked with 5% BSA for 1 h. DsRNA (i.e., replicating DENV genome) was tagged using J2 mouse monoclonal antibody mAb (Scicons, Hungary) at 1:500 dilution (1 h, 37 °C). Cells were incubated with goat-anti-mouse IgG conjugated with Alexafluor 647 (Life Technologies, Carlsbad, CA, USA) at 1:1000 dilution (1 h, 37 °C). Cells were washed with PBS between antibody incubations and finally mounted in ProLong^TM^ Gold anti-fade mountant (Molecular Probes, Eugene, OR, USA).

### 3.4. Live-Cell Fluorescence Imaging

All imaging was done using Olympus Ix83 microscope set at 37 °C, 5% CO_2_. The plate was stabilized inside the live-cell imaging chamber for 1 h prior to start of image acquisition using Hamamatsu SCMOS V3 camera. Images were acquired using the Olympus CellSens^©^ Dimension software, 60× oil immersion objective lens (N.A. 1.4), and XCite LED light source set at 30% power. For visualization, fluorescence images were deconvoluted using the built-in 2D CI-Deconvolution algorithm for widefield fluorescence in CellSens^©^ Dimension program (1 iteration, 512-pixel tile overlap). Prior to live-cell imaging at various time points, cells were washed with PBS and replaced with fresh culture media.

### 3.5. Viral Replication Assays

Viral replication kinetics after PPMO addition was determined with qRT-PCR as described elsewhere [49]. Briefly, viral RNA was extracted from the infected cells using QIAamp RNEasy mini kit (Qiagen, Hilden, Germany). The amount of viral RNA in cells was determined by using LunaScript RT one-step qRT-PCR kit (New England Biolabs, Ipswich, MA, USA). Absolute quantitation was performed by comparison of C_T_ values with a standard curve generated from known concentrations of in vitro transcribed RNA. Primer sequences are described here [49].

### 3.6. Image Analysis and Statistical Comparisons

All image analyses were performed in ImageJ software, using the OlympusViewer plugin (NIH, MA, USA). A region-of-interest (ROI) was drawn around a cell, excluding the nucleus. The nucleus was excluded since viral replication is known to occur only in the cytoplasm. A similar ROI was drawn outside of the cells that corresponds to the “background.” For each analysis, 50–100 ROIs were used. The corrected total cell fluorescence (CTCF) values were calculated using the formula:CTCF = integrated density in ROI − (area of ROI × mean intensity of “background”)(1)

The values for integrated density, area, and mean background density were obtained from the “measurement” function in ImageJ. The full procedure for determining CTCF values is detailed elsewhere [50].

Signal colocalization analysis was performed in ROIs using the *coloc2* and *coloc threshold* functions inbuilt in ImageJ. These functions include the thresholded Manders colocalization coefficient (tM1) as output [40,51,52]. The tM1 value describes the degree of colocalization of signals from channel 1 with signals from channel 2. For this analysis, 80–100 cells were used per sample.

Statistical analyses were performed using GraphPad Prism ver. 8.4.2 (San Diego, CA, USA). Means between two groups were compared using an unpaired *t*-test. Means between more than two groups were compared using two-way ANOVA.

## Figures and Tables

**Figure 1 ijms-21-09260-f001:**
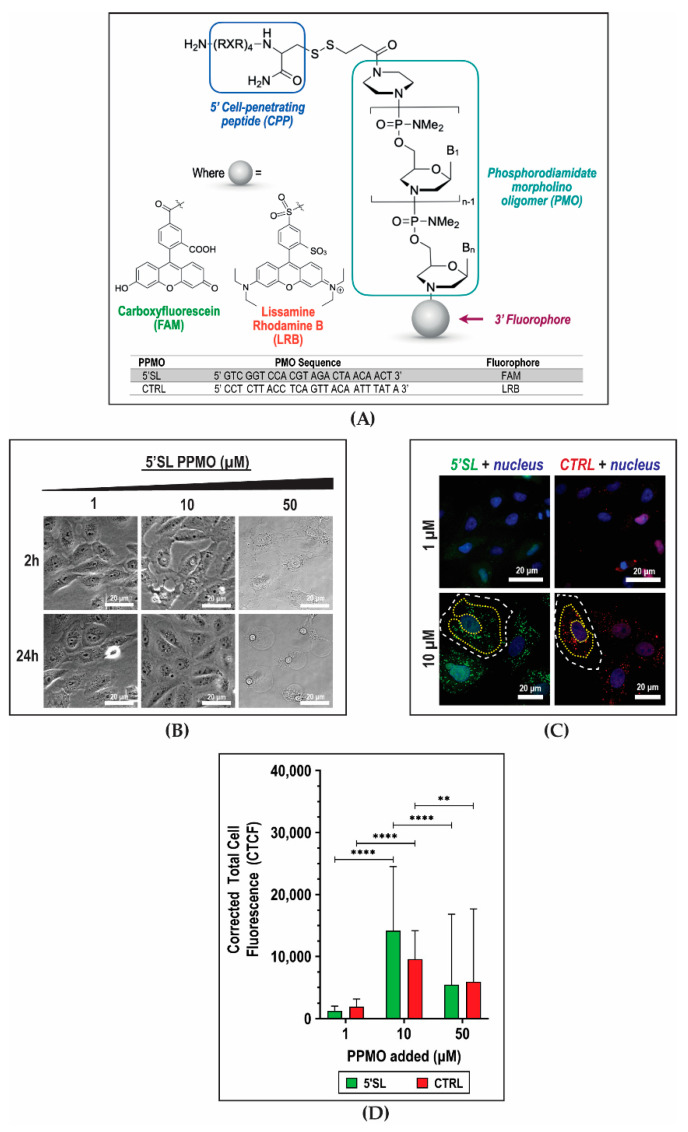
Peptide-conjugated phosphorodiamidate morpholino oligomers (PPMOs) as fluorescence imaging probes. (**A**) Schematic diagram of PPMO structure. Within the phosphorodiamidate morpholino oligomer (PMO) structure, the nuclease-resistant phosphorodiamidate bond is shown inside the bracket. The cell-penetrating peptide (CPP) (RXR)_4_B is attached to the 5′-end of the PMO, while a fluorophore is conjugated at the 3′-end. The DENV2-targeted 5′SL PPMO is tagged with carboxyfluorescein (FAM), and the non-targeted unrelated control (CTRL) PPMO is tagged with lissamine rhodamine B (LRB). (**B**–**D**) Effect of PPMO concentration on viability and imaging contrast in monkey kidney Vero cells. Representative (**B**) phase-contrast and (**C**) deconvoluted fluorescence images are shown, taken at 2 and 24 h, and 24 h, respectively. Cells are incubated for 10 min with different concentrations of PPMOs. (**D**) Quantification of fluorescence signals in cells incubated with different PPMO concentrations. The corrected total cell fluorescence (CTCF) is calculated by subtracting the contribution of background from the integrated fluorescence density within the regions of interest (ROIs) drawn around cells. The cell boundary is outlined in white dashed lines, and sample ROIs used in fluorescence signal quantification are outlined in yellow dotted lines. Columns represent mean values, and error bars represent the standard deviation. An unpaired *t*-test is used to compare means. **, *p* <0.005; ****, *p* < 0.0001.

**Figure 2 ijms-21-09260-f002:**
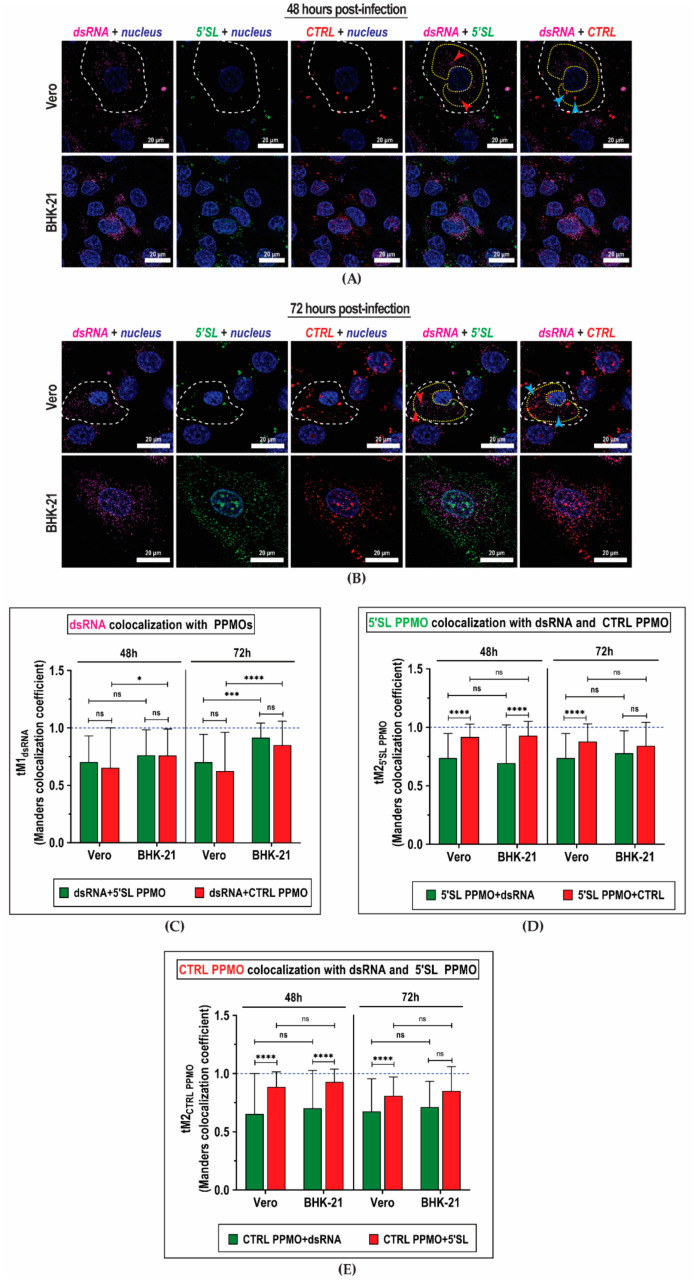
Colocalization of PPMOs with Dengue virus replication complex in vesicle packets. (**A**,**B**) Representative deconvoluted fluorescence images of monkey kidney (Vero) and baby hamster kidney (BHK-21) cells infected with 1 MOI Dengue virus serotype-2 (DENV2). Cells incubated with both targeted 5′SL and non-targeted CTRL PPMOs (10 µM final concentration each) at 1 h post-infection are subjected to immunostaining with antibody against double-stranded RNA (dsRNA) at (**A**) 48 h and (**B**) 72 h post-infection. The anti-dsRNA antibody tags virus replication complexes within cellular vesicle packets. (**C**–**E**) Analysis of colocalization of fluorescence signals by thresholded Manders coefficient (tM), which represents the degree of colocalization of two fluorescence signals. tM values are determined within regions of interest (ROIs) in fluorescence images. (**C**) Colocalization of viral replication vesicle packets with either 5′SL PPMO or CTRL PPMO evaluated at 48 h and 72 h after infection. (**D**,**E**) Colocalization of (**D**) 5′SL PPMO and (**E**) CTRL PPMO with either dsRNA or the other PPMO. The cell boundary is outlined in white dashed lines, and sample regions of interest (ROIs) used in fluorescence signal quantification are outlined in yellow dotted lines. Areas where dsRNA signals colocalize with 5′SL PPMO are indicated by red arrows, while areas where dsRNA colocalize with CTRL PPMO are indicated by blue arrows. Columns represent mean values, and error bars represent standard deviation. The blue dashed line indicates ratio = 1. Mean values are compared with two-way ANOVA. *, *p* <0.05; ***, *p* < 0.0005; ****, *p* < 0.0001; ns, not significant.

**Figure 3 ijms-21-09260-f003:**
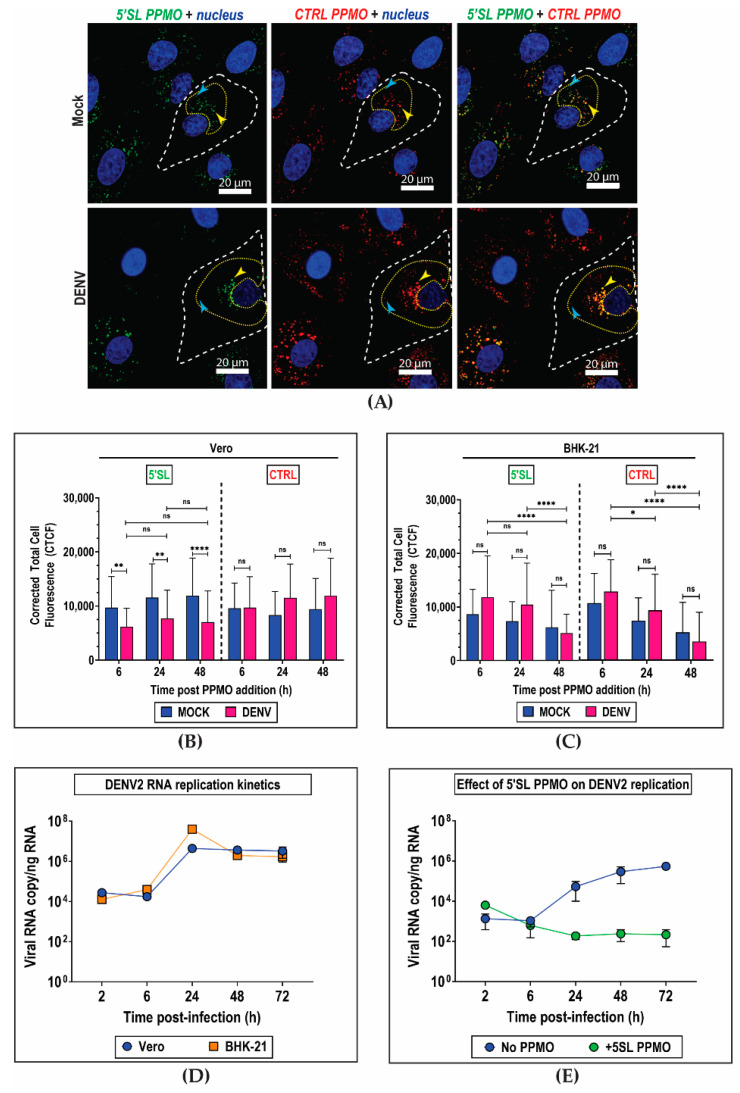
Live-cell monitoring of Dengue virus infection with PPMOs. (**A**) Representative deconvoluted fluorescent images of live monkey kidney (Vero) cells at 6 h post-infection. Cells are infected with 1 MOI Dengue virus serotype-2 (DENV2) for 1 h and immediately pulsed with either DENV2-targeted 5′SL and non-targeted CTRL PPMOs (10 µM final concentration, 10 min). The cell boundary is outlined in white dashed lines, and sample regions of interest (ROIs) used in fluorescence signal quantification are outlined in yellow dotted lines. Sample foci where the PPMOs colocalize are indicated by the yellow arrows, and where the PPMOs do not colocalize are indicated by the blue arrows. (**B**,**C**) Quantification of fluorescence signals over time within DENV2-infected and mock-infected (**B**) Vero and (**C**) baby hamster kidney (BHK-21) cells incubated with both 5′SL and CTRL PPMOs. The Corrected Total Cell Fluorescence (CTCF) is calculated by subtracting the contribution of background intensity from the integrated fluorescence density within the ROIs. (**D**) DENV2 replication kinetics in Vero and BHK-21 cells infected with 1 MOI virus (*n* = 6). (**E**) DENV2 replication kinetics in Vero cells infected with 1 MOI virus in the presence of 10 µM 5′SL PPMO (*n* = 6). Cells are pulsed with PPMOs at 1 h post-infection. Mean values in (B–C) are compared with two-way ANOVA. *, *p* < 0.05; **, *p* <0.005; ****, *p* < 0.0001; ns, not significant.

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
