# Peer review of "Peptide-Conjugated Phosphorodiamidate Morpholino Oligomers for In Situ Live-Cell Molecular Imaging of Dengue Virus Replication"

_ijms, 2020, doi:10.3390/ijms21239260_

Round 1

Reviewer 1 Report

  • The work of Ref. 15, has never been translated into a clinical practice.
  • The possibility of using this PPMO has been frequently stated for invivo PET imaging. Authors should remember that once this PPMO is labeled with a radionuclide it may compromise its current chemical/biological activities.
  • If PPMO concentration is given in M then the volume added should be state. 10µM is not a quantity.
  • A few grammatical errors should be eliminated.
  • Manuscript should be rewritten with a fewer use of acronyms.

Author Response

"Please see attachment"

Reviewer 2 Report

In this article by Bianca Leuna Victorio et al, the authors make a  fine study and describe their results of the challenges to the application of Morpholino oligomers (PPMOs) to track Dengue Virus replication. The manuscript gives an easily understandable and detailed introduction to the existing systems and tests the ability to use PPMOs as viable tracking agents for DENV in living cells. The experiments suggest that PPMOs are not deliverable sufficiently into the cells, due to the route of entry via endocytosis and discusses that this challenge should be overcome for virus tacking applications. The PPMOs while specifically inhibited DENV replication in target cells at 10uM, were not traceably specific to the DENV vRNA, behaved similar to control morpholinos exhibiting intra-cellular punctate appearance plausibly daily to exit endosomal compartments. The results, while point towards a negative outcome, are much needed to be published for the general interest of the scientific community. One thing I would like the authors to state clearly in the discussion is that the current limitation of PPMOs as imaging agents only applies to live-cell imaging, they might as well perform extraordinarily under fixed cell conditions where cell membranes are permeabilized. Perhaps even in live cells while delivering them through transient membrane permeabilization and cellular recovery, which is an involved process. Overall, the paper is well written and the shortcomings are clearly discussed. 
